# Probiotic Supplementation for Promotion of Growth in Children: A Systematic Review and Meta-Analysis

**DOI:** 10.3390/nu14010083

**Published:** 2021-12-25

**Authors:** Joseph Catania, Natasha G. Pandit, Julie M. Ehrlich, Muizz Zaman, Elizabeth Stone, Courtney Franceschi, Abigail Smith, Emily Tanner-Smith, Joseph P. Zackular, Zulfiqar A. Bhutta, Aamer Imdad

**Affiliations:** 1College of Medicine, SUNY Upstate Medical University, Syracuse, NY 13210, USA; cataniaj@upstate.edu (J.C.); panditn@upstate.edu (N.G.P.); ehrlichj@upstate.edu (J.M.E.); zamanm@upstate.edu (M.Z.); estone121@gmail.com (E.S.); cfranceschi94@gmail.com (C.F.); 2Health Science Library, SUNY Upstate Medical University, Syracuse, NY 13210, USA; smithab@upstate.edu; 3College of Education, University of Oregon, Eugene, OR 97403, USA; etanners@uoregon.edu; 4Department of Pathology, Children’s Hospital of Philadelphia, University of Pennsylvania, Philadelphia, PA 19104, USA; Joseph.Zackular@pennmedicine.upenn.edu; 5Centre for Global Child Health, The Hospital for Sick Children, Toronto, ON M5G 0A4, Canada; zulfiqar.bhutta@aku.edu; 6Center of Excellence in Women and Child Health, The Aga Khan University, Karachi 74800, Pakistan; 7Department of Pediatrics, Division of Pediatric Gastroenterology, Hepatology and Nutrition, SUNY Upstate Medical University, Syracuse, NY 13210, USA

**Keywords:** probiotics, synbiotics, growth, low- and middle-income countries, high-income countries, systematic review

## Abstract

Probiotics are commonly prescribed to promote a healthy gut microbiome in children. Our objective was to investigate the effects of probiotic supplementation on growth outcomes in children 0–59 months of age. We conducted a systematic review and meta-analysis which included randomized controlled trials (RCTs) that administered probiotics to children aged 0–59 months, with growth outcomes as a result. We completed a random-effects meta-analysis and calculated a pooled standardized mean difference (SMD) or relative risk (RR) and reported with a 95% confidence interval (CI). We included 79 RCTs, 54 from high-income countries (HIC), and 25 from low- and middle-income countries (LMIC). LMIC data showed that probiotics may have a small effect on weight (SMD: 0.26, 95% CI: 0.11–0.42, grade-certainty = low) and height (SMD 0.16, 95% CI: 0.06–0.25, grade-certainty = moderate). HIC data did not show any clinically meaningful effect on weight (SMD: 0.01, 95% CI: −0.04–0.05, grade-certainty = moderate), or height (SMD: −0.01, 95% CI: −0.06–0.04, grade-certainty = moderate). There was no evidence that probiotics affected the risk of adverse events. We conclude that in otherwise healthy children aged 0–59 months, probiotics may have a small but heterogenous effect on weight and height in LMIC but not in children from HIC.

## 1. Introduction

The role of gut microbiota in human health has been studied extensively in the recent past [1]. Observational and experimental studies from community settings have shown that gut microbiota immaturity or dysbiosis is associated with risk of development of acute malnutrition and linear growth failure in children [2,3,4,5]. Probiotics are one of the commonly used products to promote healthy gut microbiomes, encompassing a USD 54 billion industry around the globe [6,7]. Probiotics are defined as live microorganisms that, when administered in adequate amounts, confer a health benefit to the host [8]. Prebiotics are a non-digestible food ingredient that benefit the host by stimulating the growth or activity of microorganisms indigenous to the human digestive tract [9]. Synbiotics are a combination of both probiotics and prebiotics [10]. Multiple studies have assessed the usage of probiotics in the context of childhood growth [11,12,13]. A systematic review by Onubi et al. assessed the effect of probiotics on growth in children in developing countries [12]. This review of 12 studies was published in 2014 and did not include studies from high-income countries and did not use a standard method to assess the overall quality of evidence. We, therefore, aimed to systematically assess up-to-date evidence on the effects of probiotics supplementation on growth outcomes in children aged 0–59 months by following the methodological guidance of the Cochrane Collaboration.

## 2. Materials and Methods

We conducted a systematic review and meta-analysis and reported our findings according to Preferred Reporting Items for Systematic Reviews and Meta-Analyses (PRISMA) guidelines. We developed a team consensus on the study questions and methods in a protocol before the start of the study and registered this protocol on PROSPERO (CRD42020154352).

We included individual- and cluster-randomized trials. Trials with multiple treatment arms were included if the only difference between the arms was probiotic usage. We included studies with children aged 0 to 59 months who were supplemented with oral probiotics or synbiotics. We excluded studies that only tested prebiotics. As our review considered both probiotics and synbiotics, the term “probiotic” hereafter refers to both probiotics and synbiotics. The studies were included irrespective of dose, dosage forms, and strain of probiotics. We included studies that had an adequate comparison group such as standard-of-care, placebo, or no-intervention groups. We excluded observational studies such as cohort studies, case–control studies, case series, and case reports. We excluded studies on children with congenital abnormalities, syndromic diagnosis, and chronic conditions such as cystic fibrosis or inflammatory bowel disease. Furthermore, we excluded studies in which the authors declared that participants were already undernourished or malnourished.

We searched multiple electronic databases, including PubMed, Embase, Cochrane Central Register of Controlled Trials, CINAHL, Scopus, and LILACS. The last date of search was 6 November 2020. The search strategy for all databases is available in the supplementary document (Text S1, supplementary document). We searched ClinicalTrials.gov for ongoing studies. We also searched the reference sections of published studies and systematic reviews. We contacted the authors in relevant fields for any new studies. If growth data were measured but not reported in the study paper, we contacted the authors for those results. Trial registries were searched and checked for unpublished data as well as newly published data. If a study was only available in a language other than English, we attempted to translate the paper. If an adequate translation was unavailable, the study was excluded.

At least two authors screened titles and abstracts using Covidence software [14], and extracted data in duplication (AI, JC, NP, JE, MZ ES, CF). We double extracted relevant data using a data collection form specifically designed for this review. Data were extracted independently into the same form. We extracted the data for study design, study setting (hospital vs. community, country, country income status), inclusion and exclusion criteria, participant characteristics (age, nutritional status, gender), and characteristics of intervention (type, strain, form, duration, frequency, dose, comparison group). We included data on all outcomes at the longest follow-up reported by individual studies. If longest follow-up was not reported, we used suitable data the study provided which was either in the form of total growth gain over the course of the study or growth gain per time. Any duplicate data were only counted once. Any disagreement among the authors during any stage of the study was resolved by discussion and review of the publication(s) with consultation of the senior author (AI).

Our primary outcomes included weight-for-age (continuous outcome, Z scores or kg) and height-for-age (continuous outcome, cm or Z scores). Growth data were used in the form of Z scores as to WHO standards or in the primary units. Secondary outcomes included weight-for-height, BMI, head circumference, and adverse events such as nausea, vomiting, diarrhea, abdominal pain, flatulence, and sepsis.

We pooled the dichotomous outcomes to obtain a summary estimate in the form of mean relative risk (RR) and reported with its 95% confidence interval (CI). We used the standardized mean difference effect size for continuous outcomes due to studies reporting data in different units (e.g., a few studies reported weight in kg and the others in Z scores) and reported the standardized mean effect with its 95% CI. We used Review Manager 5.4 and Stata to conduct the meta-analysis [15,16]. We used the random-effects models to pool data as the effect of probiotics could be different in different study populations. We used funnel plots and Egger’s test to assess for publication bias. 

For eligible studies with multiple treatment arms, one eligible pair was selected and included, and if more than two groups were eligible, they were combined into a single pairwise comparison. If a trial had multiple arms that addressed different doses, those arms were combined and compared with the control arm to avoid double counting the control arm in the synthesis. Cluster trials were synthesized together with individually randomized trials using cluster adjusted values. If trial results were not cluster adjusted, we adjusted the result by methods given in the Cochrane handbook [17]. Extended details of data input are found in the supplementary document (Text S2). 

We assessed clinical, methodological, and statistical heterogeneity of effects reported in the literature. Statistical heterogeneity was assessed using the χ^2^ test, I^2^, and tau-squared statistics, and visual inspection of the forest plot. We considered statistical heterogeneity to be significant if the *p* value of the χ^2^ was <0.1, I^2^ values were above 50%, and forest plots showed different magnitude and effect of the intervention. We conducted subgroup analyses to explore reasons for any substantial statistical heterogeneity.

We conducted all analyses for low- and middle-income countries (LMIC) separately from high-income countries (HIC), given presumed heterogeneity in study populations across these settings. A subgroup analysis was conducted for growth outcomes using a χ^2^ test to assess whether the effects of probiotics were significantly different for the following subgroups: age: 0–<6 months vs. 6–<60 months, probiotic interventions with single vs. multiple strains, synbiotics vs. probiotics, and participant status of healthy vs. premature/low birth weight. The healthy group included participants not defined as premature/low birth weight. As previously described, we excluded all studies in which authors declared that participants were undernourished or malnourished. We also completed a post hoc analysis in which we calculated the effect of each strain, or combination of strains, for each of our main outcomes. 

We conducted sensitivity analyses by excluding studies with high risk of bias, those studies where data were supplied in a form other than mean (SD), or when effect-size data were extracted from figures. 

Two authors assessed and agreed upon bias using the Cochrane Collaboration’s risk-of-bias tool-2 (ROB-2) for assessing risk of bias for all outcomes from the included studies in the meta-analysis [18]. Using this tool, results for each outcome were judged as either low, some concerns, or high risk of bias. The certainty of overall evidence for the effect of probiotics for an outcome was assessed using the Grading of Recommendations Assessment, Development and Evaluation (GRADE) method [19]. We present the results of the quality assessment in the form of a summary-of-findings table separately for high-income countries and low- and middle-income countries. 

## 3. Results

### 3.1. Literature Search 

Our literature search identified 11,158 titles after exclusion of duplicates. Figure 1 shows the results of the literature search. The 11,158 studies were reduced to 243 full-text studies after applying the exclusion criteria stated in our methods section. After screening the full text of 243 studies, we ultimately included 79 studies in our systematic review, of which 54 studies were from high-income countries, and 25 studies from low- and middle-income countries [20,21,22,23,24,25,26,27,28,29,30,31,32,33,34,35,36,37,38,39,40,41,42,43,44,45,46,47,48,49,50,51,52,53,54,55,56,57,58,59,60,61,62,63,64,65,66,67,68,69,70,71,72,73,74,75,76,77,78,79,80,81,82,83,84,85,86,87,88,89,90,91,92,93,94,95,96,97,98]. We excluded 164 studies, and reasons for exclusion can be found in Appendix A. 

### 3.2. Characteristics of Included Studies

Appendix A display participant characteristics and intervention characteristics, respectively. The included studies had 12,524 total participants from high-income countries and 13,037 total participants from low- and middle-income countries. The median sample size for included studies was 149 with a range of 4541 (min: 15, max: 4556). Seventy-six studies were individually randomized [20,21,22,23,24,25,26,27,28,29,30,31,32,33,34,35,36,37,38,39,40,41,42,43,44,45,46,47,48,49,50,51,52,53,54,55,56,57,58,59,60,61,62,63,64,65,66,67,68,69,70,71,72,73,74,75,77,78,79,80,82,83,84,85,86,87,88,89,90,91,92,94,95,96,97,98], and three studies were cluster-randomized and were already cluster-adjusted [76,81,93]. Thirty-six of the studies were conducted in the community setting [21,22,24,25,27,28,30,37,43,45,48,52,53,54,56,58,60,63,64,67,70,71,73,74,76,77,78,80,86,87,88,89,93,98], and 43 were conducted in a hospital setting [20,23,26,29,31,32,33,34,35,36,38,39,40,41,42,44,46,47,50,51,55,57,59,61,62,65,66,68,69,72,75,79,82,83,84,85,90,91,92,95,96,97]. A total of 35 countries were represented in our meta-analysis and most of the studies were conducted in the United States [23,30,43,67,71,74,82,94,98] (more details in Text S3 in Supplementary document). Twelve studies had multiple intervention arms that we combined to obtain a single pairwise comparison [21,31,37,48,54,56,58,67,82,88,89]. Two studies were used as two separate datasets as they contained two independent treatment and control arms [33,84]. Twenty-one studies used an intervention that consisted of synbiotics [20,22,31,33,39,42,43,49,60,61,63,64,66,70,73,78,82,87,93,96,98]. Fifty-two studies were conducted on apparently healthy participants [20,21,22,24,25,27,28,30,31,33,37,40,41,43,45,46,48,49,50,52,53,54,55,56,58,60,61,62,63,64,66,67,70,71,73,74,76,77,78,79,80,83,85,86,87,88,89,91,93,94,96,98], and twenty-seven studies were conducted on premature or low birth-weight infants [23,26,29,32,34,35,36,38,39,42,44,47,51,57,59,65,68,69,72,75,81,82,84,90,92,95,97]. Thirty studies compared a probiotic intervention to a placebo [21,23,24,25,32,34,35,37,39,44,45,46,47,49,52,56,59,60,61,65,72,77,79,81,82,84,90,91,97,98], forty-eight studies compared to standard of care [20,22,26,27,28,29,30,31,33,36,38,40,41,42,43,48,50,51,53,54,55,57,58,62,63,64,66,67,68,69,70,71,73,74,75,76,78,80,83,85,86,87,88,89,93,94,95,96], and one study compared to no intervention [92]. The most common single-strain intervention used was *Bifidobacterium lactis* [33,39,59,64,70,73,75,83,85,94]. The median dose was 1.0 × 10^9^ CFUs (range 1.0 × 10^6^–1.8 × 10^10^) administered per day of study. The median duration of intervention was 13 weeks (range 1–104 weeks) for studies that reported an average duration of intervention. Thirty-two studies had a probiotic intervention that consisted of multiple strains [21,23,24,26,28,29,31,37,42,44,47,48,49,51,54,56,58,65,66,67,68,80,82,84,87,88,89,91,93,95,96,98], forty-six studies had a probiotic intervention that consisted of a single strain [20,22,25,30,32,33,34,35,36,38,39,40,41,43,45,46,50,52,53,55,57,59,60,61,62,63,64,69,70,71,72,73,74,75,76,77,78,79,81,83,85,86,90,92,94,97], and one study did not specify the probiotic content of the intervention. A total of 46 studies received industry funding [22,24,28,30,31,33,40,41,43,44,45,46,50,52,53,54,55,56,57,59,62,63,64,67,70,71,72,73,74,75,76,78,80,81,83,84,85,86,87,88,89,90,93,96,97,98]. At least one growth outcome was listed as a primary outcome in 38 studies [20,22,23,27,30,31,33,34,36,40,41,42,43,44,52,54,56,58,59,63,66,67,70,71,74,77,78,82,83,84,85,86,87,88,92,94,96,97].

### 3.3. LMIC Results

#### 3.3.1. Weight-for-Age

In LMIC, twenty-one studies reported data on weight-for-age and included a total of 8417 participants (4323 probiotics, 4094 control) [21,26,27,33,36,38,39,50,58,60,61,68,69,70,72,77,83,84,85,91,92]. The results showed low-certainty evidence that probiotics had a small effect on weight when compared to the control group (SMD: 0.26, 95% CI: 0.11–0.42, *p* = 0.001, I^2^ = 87%, Figure 2). The GRADE evidence was downgraded to low due to statistical heterogeneity in the pooled data and clinical heterogeneity in the use of probiotics in the included studies (Table 1). The funnel plot and Egger’s test showed no evidence of publication bias (*p* = 0.07, Funnel plot in Appendix A). The summary risk of bias for weight-for-age in low- and middle-income countries is shown in Figure 3. The results were similar without high-risk-of-bias studies (SMD 0.31, 95% CI: 0.13–0.48, *p* < 0.001, I^2^ = 89%). A subgroup analysis comparing single versus multiple strain interventions found that single-strain interventions had a greater impact on weight gain than multiple strain interventions (P_subgroup_ = 0.002). Figure 4 shows the results of other sensitivity analyses and subgroup analyses. All other subgroup and sensitivity analyses did not show significantly different results for weight-for-age. A post hoc subgroup analysis based on type of strains did not show a particular individual strain or combination of strains that were effective in terms of effect on weight (Appendix A).

#### 3.3.2. Height-for-Age

Twelve studies from LMIC reported data on height-for-age and included a total of 2561 participants (1347 probiotics, 1214 control) [21,27,33,36,39,50,70,83,84,85,91,92]. The results provided moderate-certainty evidence that probiotics had a small effect on height-for-age when compared to the control group (SMD: 0.16, 95% CI: 0.06–0.25, *p* = 0.002, I^2^ = 25%, Figure 5). The GRADE evidence was downgraded due to heterogeneity in the clinical use of probiotics in the included studies (Table 1). The funnel plot and Egger’s test showed no evidence of publication bias (*p* = 0.25). All subgroup and sensitivity analyses did not show significantly different results for height-for-age (Figure 4). A summary risk of bias for height-for-age and other outcomes in LMIC are available in Appendix A.

#### 3.3.3. Other Outcomes

Among studies from LMIC, probiotics did not have any significant effect on other outcomes including adverse events (forest plots in Appendix A). A funnel plot for head circumference in LMIC is available in Appendix A. The other analyses for LMIC, including subgroup and sensitivity analyses, are displayed in Figure 4.

### 3.4. HIC Results

#### 3.4.1. Weight-for-Age

Fifty-one studies from HIC reported data on weight-for-age and included a total of 10,832 participants (5759 probiotics, 5074 control) [20,22,23,24,25,28,29,30,31,32,34,35,37,40,41,44,45,46,47,48,49,51,53,54,55,56,57,59,62,63,64,65,66,67,71,73,74,75,78,79,80,81,82,86,87,88,89,90,94,96,97]. The results provided moderate-certainty evidence that probiotics did not have a clinically meaningful effect on weight when compared to the control group (SMD: 0.01, 95% CI: −0.04–0.05, *p* = 0.78, I^2^ = 7%, Appendix A). The GRADE evidence was downgraded due to clinical heterogeneity in the use of probiotics in the included studies (Table 2). The funnel plot and Egger’s test showed no evidence of publication bias (*p* = 0.64, Appendix A). The sensitivity analysis without high-risk-of-bias studies showed similar results (SMD 0.00, 95% CI: −0.04–0.05, *p* = 0.88, I^2^ = 6%). A post hoc subgroup analysis based on type of strain did not show a strain or combination of strains that influenced any of the growth outcomes from studies from high-income countries (Appendix A). All subgroup and sensitivity analyses did not show significantly different results for weight-for-age (Appendix A).

#### 3.4.2. Height-for-Age

Thirty-two studies from HIC reported data on height-for-age and included a total of 6118 participants(3350 probiotics, 2768 control) [20,22,25,28,30,31,37,40,41,44,45,49,53,54,55,56,63,66,67,71,73,75,78,79,80,86,87,88,90,94,96,97]. The results provided moderate evidence that probiotics did not have a clinically meaningful effect on height-for-age when compared to the control group (SMD: −0.01, 95% CI: −0.06–0.04, *p* = 0.71, I^2^ = 0%, Appendix A). The GRADE evidence was downgraded due to clinical heterogeneity of probiotics used in the included studies. The funnel plot and Egger’s test showed no evidence of publication bias (*p* = 0.87, Appendix A). All subgroup and sensitivity analyses did not show significantly different results for height-for-age (Appendix A).

#### 3.4.3. Other Outcomes

Probiotics did not show any significant effect on other primary and secondary outcomes in studies from HIC (forest plots in Appendix A). The summary risk of bias for effect of probiotics for weight-for-age and other outcomes is shown in Appendix A. A funnel plot for head circumference in HIC is found in Appendix A. The other analyses for HIC, including subgroup and sensitivity analyses, are displayed in Appendix A.

## 4. Discussion

This comprehensive systematic review evaluated the effects of probiotics on growth in children 0 to 59 months of age. Overall, there was no evidence that probiotics had a clinically meaningful effect on any of the growth outcomes in children from high-income countries. The data from low- and middle-income countries showed that there may be a small beneficial effect on weight and height gain; however, the certainty of evidence was low and moderate for these outcomes. There was no evidence that probiotics increased the risk of any of the adverse events including risk of sepsis from HIC and LMIC.

We used the GRADE approach to assess the overall certainty of evidence for the effect of probiotics on primary outcomes and selected secondary outcomes. The GRADE method of certainty assessment gives ratings of evidence for each outcome and considers factors such as type of study, risk of bias, inconsistency of results, indirectness of evidence, imprecision of the summary estimate, and publication bias [19]. All included studies were randomized trials, and we did not downgrade the certainty rating for study designs for any of the outcomes graded. Overall, there were a few studies that were at high risk of bias, and sensitivity analyses removing these studies did not substantively change the results, so we did not adjust the overall certainty of evidence rating for risk of bias. However, we did adjust the certainty rating due to clinical heterogeneity in the use of probiotics for all the outcomes graded. We conducted separate GRADE assessments for studies from high-income countries and those from low- and middle-income countries because we believe that environmental factors, diet, and gut microbiome might be different in these settings [2]. We therefore did not downgrade the certainty of evidence for indirectness for any of the graded outcomes. The number of studies that contributed data for an outcome varied among the outcomes. We downgraded the evidence for imprecision where the number of included studies was small and the confidence interval included a null effect.

Even though there was clinical heterogeneity in the use of probiotics, the pooled results were mostly homogenous around the null effect, especially from studies from high-income countries. Therefore, it can be concluded with reasonable confidence that probiotic supplementation in otherwise healthy children from high-income countries does not give any differential effect in terms of growth. The findings from trials conducted in low- and middle-income countries were mixed. The pooled results for the effects of probiotics on weight-for-age from low- and middle-income countries showed a small effect in favor of the intervention (SMD 0.26, 95% CI 0.11–0.42); however, there was significant statistical heterogeneity in the pooled data (I^2^ = 87%) and the positive effect can be explained by three studies [27,36,92] that had an effect SMD > 1. Removal of these studies with outlying findings yielded results that were similar to those from high-income countries (SMD 0.05, 95% CI −0.02, 0.12). We adjusted the GRADE evidence by lowering the certainty grade to ‘low’ for this outcome, which means that we have low confidence in this estimate and future research might change this estimate.

Were there any subgroups that might be affected differently? The data from high-income countries did not show any differential effects of probiotics for groups such as age < 6 months vs. 6–59 months, probiotics vs. synbiotics, or single vs. multiple strains and nutritional status (Appendix A). Thus, there does not seem to be an overall effect or any significant effect in subgroups from studies from high-income countries. The subgroup analyses from low- and middle-income countries are hard to interpret as the number of studies in each subgroup varied and some of the differences observed could be due to the small number of included studies. Nonetheless, there was evidence that single-strain probiotics may yield a more pronounced effect on weight-for-age compared to multiple-strain probiotics (Figure 4), although no particular single strain can be attributed to this result. This result was only present when pooling many studies that utilized a single-strain intervention. Future large studies will be required to further elaborate on and potentially replicate this finding. Indeed, more-targeted therapies could be more beneficial rather than using a single strain or combination of strains of probiotics [3].

This study is one of the largest systematic reviews conducted on the subject. We searched multiple databases and examined 11,158 titles and abstracts, and this included both published and ongoing studies. We did not apply any limitations of the literature search and did not exclude studies at the title/abstract screenings stages if they did not report outcomes in the abstract. This might be the reason that we were able to include many more studies compared to the last review published in 2014 [12]. We specified our analyses a priori and registered our protocol on a publicly available website before the review started. The major post hoc decision was to include studies from high-income countries as we understand that probiotics are commonly used to promote healthy gut microbiome in high-income countries, and it is important to review the available literature for their effect in promotion of healthy growth in children. We therefore conducted separate meta-analyses and certainty ratings for high-income countries and low- and middle-income countries. The limitations of the data presented in this study were that there was significant clinical heterogeneity in the use of probiotics with respect to type, duration, and combination. It is debatable if a meta-analysis should have been performed in the presence of such clinical heterogeneity. We believe that a meta-analysis was appropriate here as it served the purpose of assessing the overall magnitude and direction of effect from included studies. We adjusted the GRADE ratings for each outcome for clinical heterogeneity and conducted a post hoc subgroup analysis examining each strain or combination of strains and found no differential effect for any single strain or combination of strains for any of the growth outcomes assessed. Another potential limitation is related to type of growth outcomes included. Although we included a range of outcomes for growth, most of the outcomes were continuous and we did not include dichotomous outcomes such as undernutrition and stunting. We posit that if probiotics had any significant effect on prevention of stunting and undernutrition then that pattern would be mirrored in their effects on average height and weight gain.

There is a growing appreciation for the role of the gut microbiota, and accumulating evidence suggests an association between immaturity of the microbial community and undernutrition [4,99,100]. Although our understanding of the dynamic interplay between undernutrition and the microbiota is improving, empirical work examining therapeutic interventions to ameliorate microbiota dysbiosis remains limited. Methods that harness the microbiota through rationally designed microbial consortia, or manipulate the community through microbiota-directed therapeutic foods, show promise for treatment [3,4]. Our observation that children in low- and middle-income countries might benefit from probiotic supplementation is consistent with recent findings regarding the beneficial effects of a microbiota-targeted food intervention in Bangladesh [3]. There is a need for large-scale clinical trials that address the multifaceted role of the microbiota in childhood nutrition.

## 5. Conclusions

Probiotic supplementation does not seem to have a clinically meaningful effect on growth for apparently healthy children in high-income countries. However, there might be a small effect on weight and height in apparently healthy children from low- and middle-income countries. Future large-scale clinical trials are needed that assess the targeted therapies to prevent the gut dysbiosis associated with childhood undernutrition in low- and middle-income countries.

## Figures and Tables

**Figure 1 nutrients-14-00083-f001:**
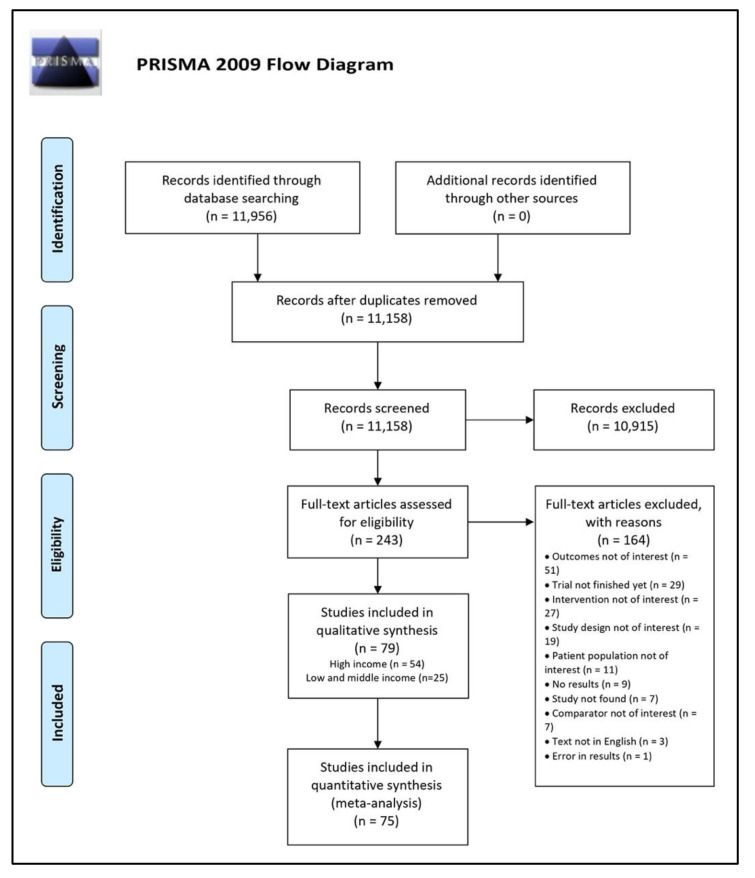
PRISMA Flow Diagram.

**Figure 2 nutrients-14-00083-f002:**
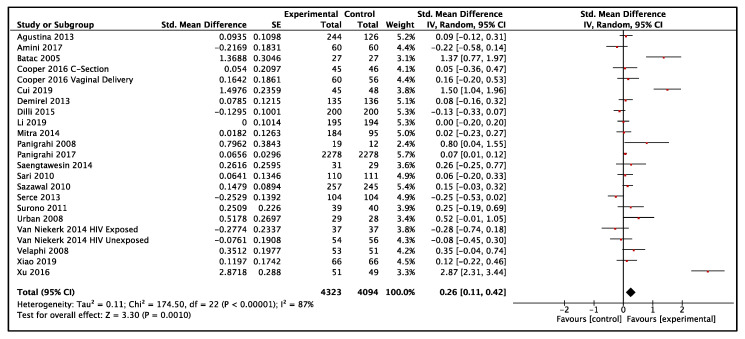
Effect of use of probiotics on weight-for-age in children 0–59 months of age from LMIC [21,26,27,33,36,38,39,50,58,60,61,68,69,70,72,77,83,84,85,91,92].

**Figure 3 nutrients-14-00083-f003:**
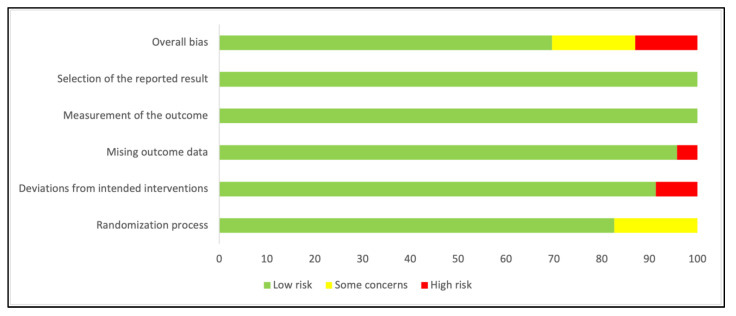
Risk of bias for weight-for-age in LMIC. The figure shows the five domains of the risk of bias 2.0 tool and the overall bias. The horizontal axis depicts percentage of studies. Most of the studies were at low risk of bias.

**Figure 4 nutrients-14-00083-f004:**
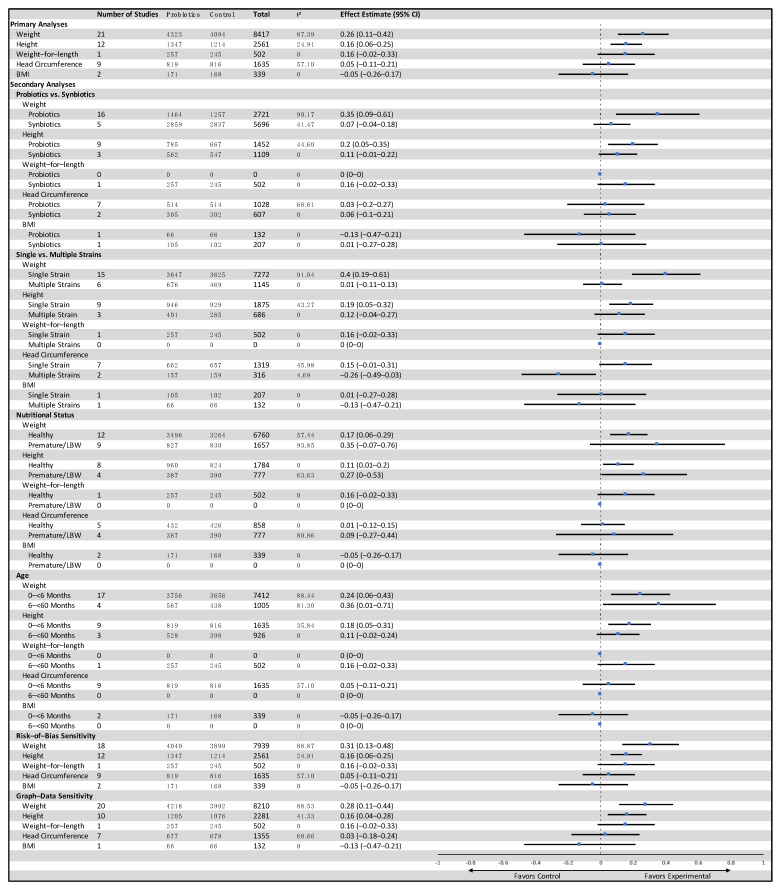
Primary and subgroup analyses for the effects of probiotics and growth outcomes in children from low- and middle-income countries.

**Figure 5 nutrients-14-00083-f005:**
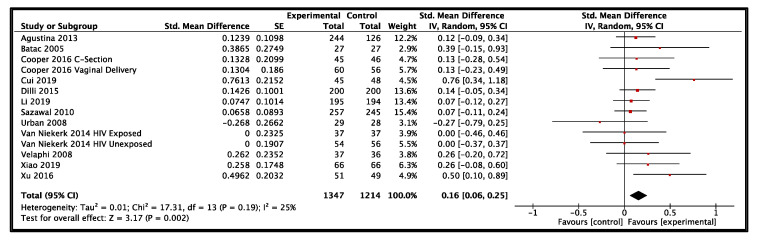
Effect of use of probiotics on height-for-age in children 0–59 months of age from low- and middle-income countries [21,27,33,36,39,50,70,83,84,85,91,92].

**Table 1 nutrients-14-00083-t001:** GRADE evidence profile showing results of GRADE analysis of overall certainty of evidence for effect of probiotics in children 0–59 months of age in low- and middle-income countries.

Certainty Assessment							№ of Patients		Effect		Certainty
No of Studies	Study Design	Risk of Bias	Inconsistency	Indirectness	Imprecision	Other Considerations	Probiotics	Control	Relative(95% CI)	Absolute (95% CI)	
Weight-for-age											
21	RCT	not serious ^a^	very serious ^b^	not serious ^c^	not serious ^d^	none	4323	4094	-	SMD 0.26 higher (0.11 higher to 0.42 higher)	⨁⨁◯◯ LOW
Height-for-age											
12	RCT	not serious ^e^	serious ^f^	not serious ^c^	not serious	none	1347	1214	-	SMD 0.16 higher (0.06 higher to 0.25 higher)	⨁⨁⨁◯ MODERATE
Head Circumference											
9	RCT	not serious ^e^	serious ^g^	not serious ^c^	serious ^h^	none	819	816	-	SMD 0.05 higher (0.11 lower to 0.21 higher)	⨁⨁◯◯ LOW
BMI											
2	RCT	not serious ^e^	serious ^i^	not serious	serious ^j^	none	171	168	-	SMD 0.05 lower (0.26 lower to 0.17 higher)	⨁⨁◯◯ LOW
Sepsis											
9	RCT	not serious ^k^	serious ^l^	not serious	not serious	none	312/3026 (10.3%)	441/3024 (14.6%)	RR 0.74 (0.64 to 0.87)	38 fewer per 1000 (from 53 fewer to 19 fewer)	⨁⨁⨁◯ MODERATE

Footnotes: ^a.^ Even though three of the included studies were at high risk of bias for this outcome, a sensitivity analysis by excluding these studies did not change the magnitude, direction, or statistical significance of the summary estimate. ^b.^ The I^2^ was 87%. Inspection of the forest plot showed the effect of probiotics varied in magnitude. We also downgraded for heterogeneity in the use of probiotics used in the included studies. ^c.^ All the studies were conducted in low-r and middle-income countries ^d.^ Overall sample size from all the included studies in the meta-analysis was more than 8000. The CI did not include 0. ^e.^ None of the included studies in this analysis were at high risk of bias. ^f.^ Even though the statistical heterogeneity was only 25%, we downgraded for clinical heterogeneity in the use of probiotics in the included studies. ^g.^ The I^2^ was 57% ^h.^ The overall sample size was less than 2000 and the confidence interval of the summary estimate included 0. ^i.^ Even though the statistical heterogeneity was only 0%, we downgraded for clinical heterogeneity in the use of probiotics in the included studies. ^j.^ The overall sample size of the included studies was less than 400 and the confidence interval of the summary estimate was wide and included 0. ^k.^ Even though one of the included studies was at high risk of bias for this outcome, a sensitivity analysis by excluding this study did not change the magnitude, direction, or statistical significance of the summary estimate. ^l.^ Even though the statistical heterogeneity was only 20%, we downgraded for clinical heterogeneity in the use of probiotics in the included studies. Abbreviations: CI: confidence interval; SMD: standardized mean difference; RR: risk ratio, BMI: body mass index RCT: randomized controlled trial.

**Table 2 nutrients-14-00083-t002:** Summary-of-Findings table showing results of GRADE analysis of overall evidence for effect of probiotics in children 0–59 months of age in high-income countries.

Certainty Assessment							№ of Patients		Effect		Certainty
No of Studies	Study Design	Risk of Bias	Inconsistency	Indirectness	Imprecision	Other Considerations	Probiotics	Control	Relative (95% CI)	Absolute (95% CI)	
Weight-for-age											
51	RCT	not serious ^a^	serious ^b^	not serious ^c^	not serious ^d^	none	5759	5073	-	SMD 0.01 higher (0.04 lower to 0.05 higher)	⨁⨁⨁◯ MODERATE
Height-for-age											
32	RCT	not serious ^a^	serious ^e^	not serious ^c^	not serious ^d^	none	3350	2768	-	SMD 0.01 lower (0.06 lower to 0.04 higher)	⨁⨁⨁◯ MODERATE
Head Circumference											
28	RCT	not serious ^f^	serious ^g^	not serious ^c^	not serious ^d^	none	2655	2117	-	SMD 0.04 lower (0.2 lower to 0.11 higher)	⨁⨁⨁◯ MODERATE
BMI											
5	RCT	not serious	serious ^e^	not serious ^c^	serious ^h^	none	415	305	-	SMD 0.09 higher (0.06 lower to 0.25 higher)	⨁⨁◯◯ LOW
Sepsis											
12	RCT	not serious ^i^	serious ^j^	not serious ^c^	serious ^k^	none	275/1778 (15.5%)	278/1749 (15.9%)	RR 1.03 (0.84 to 1.26)	5 more per 1000 (from 25 fewer to 41 more)	⨁⨁◯◯ LOW

Explanations: ^a.^ Even though four of the included studies were at high risk of bias for this outcome, the sensitivity analysis by the exclusion of these studies did not change the magnitude, direction, or statistical significance of the analysis. ^b.^ Even though the statistical heterogeneity was very low with I^2^ values of 7%, we downgraded for clinical heterogeneity in the type of probiotics used in the included studies. ^c.^ All the studies were conducted in high-income countries. ^d.^ The confidence interval of the effect size includes 0, and we think the overall effect is about 0. The confidence intervals are narrow enough that we do not think that the summary size is imprecise. ^e.^ Even though the statistical heterogeneity was homogenous with I^2^ values of 0%, we downgraded for clinical heterogeneity in the type of probiotics used in the included studies. ^f.^ Even though three of the included studies were at high risk of bias for this outcome, the sensitivity analysis by the exclusion of these studies did not change the magnitude, direction, or statistical significance of the analysis. ^g.^ The I^2^ was 85% ^h.^ The total sample size of all the studies included in the meta-analysis was less than 1000. The CIs were wide ^i.^ Even though one of the included studies was at high risk of bias for this outcome, the sensitivity analysis by the exclusion of this study did not change the magnitude, direction, or statistical significance of the analysis. ^j.^ Even though the statistical heterogeneity was low with I^2^ values of 26%, we downgraded for clinical heterogeneity in the type of probiotics used in the included studies. ^k.^ The total sample size of all the studies included in the meta-analysis was less than 1000. The CIs were wide, and increased risk cannot be excluded. Abbreviations: CI: confidence interval; SMD: standardized mean difference; RR: risk ratio, BMI: body mass index RCT: randomized controlled trial.

## Data Availability

We will share data-extraction sheets. Furthermore, we are willing to share our risk-of-bias assessment and meta-analysis RevMan file on request via email.

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
