# Peer review of "Probiotic Supplementation for Promotion of Growth in Children: A Systematic Review and Meta-Analysis"

_nutrients, 2021, doi:10.3390/nu14010083_

Round 1

Reviewer 1 Report

This meta-analysis study aimed to investigate the effects of probiotic supplementation on growth outcomes in children 0-59 months of age. This is an interesting topic especially given that probiotics are commonly prescribed to promote a healthy gut microbiome in children. The paper is well written. A few comments need to be addressed.

  1. Is it possible to conduct subgroup analysis stratified by intervention dose, intervention time, and Individual and cluster randomized trials?
  2. In figure 1, it is inappropriate to use the word “wrong”. Using “not of interest” might be better.
  3. How to define or assess significantly different results between subgroups.

Reviewer 2 Report

I read the review "Probiotic supplementation for promotion of growth in children: a systematic review and meta-analysis" very carefully. It is an interesting review of the data in the literature considering only studies in which probiotics were used. The data collected highlight the lack of positive effects of probiotics of growth children, in accordance with the belief, mostly reported, according to which probiotics could only have beneficial effects for the treatment or prevention of pathological conditions.
